# Supramolecular Host–Guest Assemblies of [M₆Cl₁₄]²⁻, M = Mo, W, Clusters with γ-Cyclodextrin for the Development of CLUSPOMs

Anton A. Ivanov [1], Pavel A. Abramov [1,*], Mohamed Haouas [2], Yann Molard [3], Stéphane Cordier [3], Clément Falaise [2], Emmanuel Cadot [2] and Michael A. Shestopalov [1]

[1] Nikolaev Institute of Inorganic Chemistry SB RAS, Novosibirsk 630090, Russia
[2] Institut Lavoisier de Versailles, UMR 8180 CNRS, UVSQ, Université Paris-Saclay, 78035 Versailles, France
[3] Université de Rennes, CNRS, ISCR—UMR 6226, ScanMAT—UAR 2025, 35000 Rennes, France
[*] Correspondence: abramov@niic.nsc.ru

**Abstract:** Host–guest assemblies open up opportunities for developing novel functional CLUSPOM multicomponent systems based on transition metal clusters (CLUS), polyoxometalates (POMs) and macrocyclic organic ligands. In water–ethanol solution γ-cyclodextrin (γ-CD) interacts with halide metal clusters [M₆Cl₁₄]²⁻ (M = Mo, W) to form sandwich-type structures. The supramolecular association between the clusters and CDs, however, remains weak in solution, and the interactions are not strong enough to prevent the hydrolysis of the inorganic guest. Although analysis of the resulting crystal structures reveals inclusion complexation, ¹H NMR experiments in solution show no specific affinity between the two components. The luminescent properties of the host–guest compounds in comparison with the initial cluster complexes are also studied to evaluate the influence of CD.

**Keywords:** molybdenum; tungsten; octahedral cluster complexes; γ-cyclodextrin; luminescence; crystal structure





## 1. Introduction

Supramolecular self-assembly systems represent an appealing way to develop and design multifunctional materials with wide variety of applications [1,2]. Among the different strategies to construct architectures built on non-covalent interactions, the host–guest chemistry is highly attractive to associate two molecular components taking benefit of their mutual recognition properties [3,4]. In this context, cyclodextrins (cyclic oligosaccharides, CDs) are highly promising macrocyclic hosts defining a hydrophobic cavity that can encapsulate various organic compounds [5–8], including drugs [9,10] which has found real applications in the food and pharmaceutical industries [11,12]. The hydrophobic effect is often pointed out as main driving forces for such purely organic inclusion complexes [13]. On the other hand, cyclodextrins are also able to bind water-soluble nanosized inorganic polyanions, such as polyoxometalates [14–16], borate clusters [17,18], or octahedral transition metal clusters [19–23]. The formation of such hybrid inorganic-organic host guest complexes offers the possibilities of tuning the photophysical characteristics [19,20], the redox properties [20], and improving the hydrolytic stability of the inorganic guests in aqueous solution [14,21,24]. Importantly, the inclusion of nanosized inorganic entities within the CD cavity is driven by the chaotropic effect that is an ion-specific effect leading to the aggregation of chaotropic ions with non-ionic organic matter, including macrocycles, surfactants, or polymers [25–27]. From a thermodynamic point of view, this solvent effect particularly exacerbated for the low-charged polyanions is orthogonal to the hydrophobic effect [28]. Indeed, this enthalpy-driven process originates from a recovery of the water network through restoration of hydrogen bonds.

The use of supramolecular approaches opens new horizons for obtaining different functional materials. For example, the combination of electron-deficient polyoxometalates

and electron-rich metal cluster complexes in one system (CLUSPOM) can lead to the creation of new photocatalysts for hydrogen evolution reactions [29–31]. Both types of compounds, as mentioned above, form strong supramolecular systems with cyclodextrins, i.e., CDs can act as binding agents between cluster and POM. To this end, studying the interaction of metal clusters with CDs is a very important task in order to limit the range of potentially promising compounds. On the other hand, such an approach is not limited to assemblies with CDs. To date, in order to eliminate the high instability of octahedral halide molybdenum clusters in water, the combination with triblock copolymers has been applied, resulting in a stable water solution of $[Mo_6I_{14}]^{2-}$ [32].

We previously reported the host–guest stabilization of halide octahedral clusters $[M_6X^i_8Cl^a_6]^{2-}$ (M = Mo, W; X = Br, I; i = inner, a = apical) in aqueous solutions by association with γ-cyclodextrin [19,21]. The formation of such complexes significantly slows down the rate of substitution of terminal chlorine ligands by water molecules. Thus, the stabilizing role of γ-CD is identified as providing a protective shell against hydrolytic attack by water. Herein, we extend the study to the series with inner chlorine ligand-containing clusters $[M_6Cl^i_8Cl^a_6]^{2-}$. Surprisingly, all data indicated no CD-stabilizing effect on these clusters, although a series of inclusion compounds built from cluster complexes $[M_6Cl^i_8Cl^a_6]^{2-}$ and γ-CD were revealed by X-ray diffraction in solid-state. The $^1$H NMR investigation confirmed the weak ability of molecular components to self-associate in water-methanol solution. Finally, we report the luminescent properties of the novel compounds.

## 2. Results and Discussion

### 2.1. Inclusion Phenomenon for CLUSPOM Materials

The ability of cyclodextrins to incorporate coordination compounds and modify their chemical and physical properties was noticed about 40 years ago. For example, the inclusion of ferrocene and its derivatives in β-cyclodextrin causes a significant positive shift of the oxidation half-wave, a slowing of the oxidation kinetics, a significant decrease in volatility, and an increase in the thermal stability of the guest [33,34]. Conversely, including the $[NiL]^{2+/+}$ pair (L = 5,7,7,12,14,14-hexamethyl-1,4,8,11-tetraazacyclotetradecane) stabilizes Ni(I) [35]. Incorporation of $[CpFe(CO)_2CH_3]$ into β- or γ-cyclodextrin changes the direction of the reactions with tributylphosphine so that only CO substitution products are formed [36]. Stabilization is particularly important for labile organometallic compounds: the inclusion of alkyl(aqua)cobaloximes $[Co(Hdmg)_2R(H_2O)]$ (Hdmg = dimethylglyoxime) into CDs (R = *n*-Pr, *n*-Bu, *i*-Bu, *n*-Am etc.) increases their thermal stability [37]. For the luminescent complexes of ruthenium $[Ru(R-phen)_3]^{2+}$ and rhenium $[Re(CO)_3L(4-R-py)]ClO_4$ (L = 2,2'-bipyridine, 1,10-Phenanthroline; R can be from $CH_3$ to $C_{13}H_{27}$), binding to β-CD increases the quantum yield, the lifetime of the excited state and prevents the quenching of the luminescence by dioxygen [38,39]. Perhaps the most important is the positive effect of host–guest complex formation on catalytic processes. The role of the CD is twofold: on the one hand, to entrap reactive substrates and carry out their interfacial transfer to the catalyst, which accelerates the process and increases the yield of the target product. The other is to entrap the catalyst (or part of it) and protect it from degradation. Both effects are manifested in various catalytic processes catalyzed by complexes of noble metals, especially with phosphines–hydroformylation, hydrogenation of aldehydes to alcohols, oxidation of alkenes to ketones, and hydrosilylation. For example, in the hydrosilylation reaction, free $[Pd(COD)Cl_2]$ (COD = 1,5-cyclooctadiene) shows no catalytic activity, which appears only when it is incorporated in β-CD [40–42].

The interaction of chalcogenide and halide cluster complexes with cyclodextrins has not been studied before our research. On the other hand, another macrocyclic cavitand, cucurbit[6]uril ($C_{36}H_{36}N_{24}O_{12}$), has been used for the crystallization of triangular cluster aqua complexes $[M_3Q_4(H_2O)_9]^{4+}$ (M = Mo, W) and their derived heterometallic cubane clusters. It has $D_{6h}$ symmetry, and the presence of a large cavity bounded by two identical "portals" allows it to be classified as a cavitand. Each portal has 6 carbonyl groups, which are geometrically well suited to form hydrogen bonds with the indicated six water molecules.

Crystallization of supramolecular compounds occurs when cucurbituril is added to a solution of the cluster aqua complex in 2–4 M HCl, already at concentrations of the aqua complex at the level of $10^{-2}$–$10^{-3}$ M. The method turned out to be universal and made it possible to isolate and structurally characterize a large number of new heterometallic clusters. In this case, two types of adducts are formed: with a cluster/cucurbituril ratio of 1:1 and 2:1 [43,44]. However, the extremely low solubility of both cucurbituril itself and especially its supramolecular adducts with clusters makes it impossible to further study the chemistry of these interesting compounds.

Another class of promising compounds from a biomedical point of view are sulfide and halide clusters. The most studied are rhenium derivatives containing an octahedral cluster core $\{Re_6S_8\}^{2+}$. They have luminescence properties in the region of the spectrum required for photodynamic therapy (PDT). The high stability of the cluster core makes such complexes attractive objects for PDT and targeted delivery of drugs and genes. The cluster complexes themselves have been shown to have low cytotoxicity [45]. The main disadvantage is the high cost of rhenium and the difficulty of synthesizing the starting compounds. However, octahedral clusters of this type can also be obtained for Nb, Ta, Mo, W, Fe, Ru, Co. The $\{Mo_6I_8\}^{4+}$ clusters have record luminescence properties, and the luminescence wavelength is in the range required for PDT [46]. By changing the terminal ligands, both hydrophilic and lipophilic complexes can be easily synthesized. Such clusters can be used as sensors and sensitizers for the conversion of triplet to singlet oxygen, including for medical purposes [32,47,48]. However, unlike rhenium clusters, molybdenum (and tungsten) iodide clusters tend to lose ligands and form insoluble hydroxides in aqueous solution, which severely limits their application. Their incorporation into cyclodextrins, in addition to the obvious effect on luminescence, redox properties, and biological activity, should significantly increase the stability of the clusters. The same is true of chalcogenide clusters of 3D transition metals.

The appeal of POMs lies in their extraordinary structural diversity and the ability to incorporate various heterometals to form even more complex structures. Such hybrid POMs have a number of unique properties. First of all, their stability at high temperatures (up to 300–400 °C) combined with their ability to accept up to 32 electrons and coordinate reactive oxygen species makes them promising oxidation catalysts [49–51]. The study of the biological activity of POM showed that they have a wide spectrum of action. Polymolybdates induce apoptosis of cancer cells. Heteropolytungstates show activity against a wide range of RNA viruses, for example, by inhibiting virus binding to the cell wall. Polyoxotungstates penetrate bacterial cell walls and increase the activity of beta-lactam antibiotics against *Staphylococcus aureus* by inhibiting the expression of the gene responsible for the production of the penicillin-binding protein. Some POMs are inhibitors of sulfotransferases. The study of the biological activity of POMs has already reached the stage of in vivo experiments [52–54]. The interaction of POMs with CDs was first mentioned by F. Stoddart et al. who described an interaction between γ-cyclodextrin molecules and the Keggin anion $[PMo_{12}O_{40}]^{3-}$ both in the solid phase and in aqueous solution [15]. Wu et al. presented molecular sieves based on cyclodextrin and POM that separate two types of CdTe nanoparticles of 3.3 and 4.4 nm size [55].

Therefore, the development of synthetic routes to combine transition metal cluster complexes and POM is of great importance. Such CLUSPOM combinations within one crystal structure will provide the following crucial features: (i) charge separation in the resting state (electron-donating cluster + electron-accepting POM), which is important for photo/electro catalysis, sensors, electronics etc.; (ii) photoactivity with the possibility of charge transfer from cluster to POM; (iii) high total number of heavy atoms which is important for bio applications (as nanoparticles), such as X-ray contrast agents, angiography etc.; (iv) photophysical properties (luminescence).

One of the most efficient strategies for combining these building blocks is to use cyclodextrin as a specific molecular "glue". For example, a three-component hydrogel material was obtained by mixing a cationic tantalum cluster $[\{Ta_6Br_{12}\}(H_2O)_6]^{2+}$,

γ-cyclodextrin, and a Dawson-type anionic POM $[P_2W_{18}O_{62}]^{6-}$ [22]. In this case, a strongly bound adduct of a cluster with CD interacts with POM to form infinite chains (alternation of a cluster and POM). A similar approach was applied to the cationic rhenium cluster $[\{Re_6Se_8\}(H_2O)_6]^{2+}$ [29], leading to the preparation of tightly packed molecular systems $[\{Re_6Se_8\}(H_2O)_6]_2\{[P_2W_{18}O_{62}]@2\gamma\text{-CD}\}^{2-}$. In addition to the intuitively obvious bindings of oppositely charged polynuclear clusters, the work demonstrated the possibility of forming a three-component system of both anionic cluster $[\{Re_6Se_8\}(CN)_6]^{4-}$ and POM $[P_2W_{18}O_{62}]^{6-}$ [29], realized using a γ-cyclodextrin linker. A similar approach was realized when the supramolecular adduct $\{[\{Re_6Te_8\}(CN)_6]@2\gamma\text{-CD}\}^{4-}$ was incorporated into the cavity of the $[Mo_{152}O_{457}H_{14}(H2O)_{68}]^{16-}$ nanowheel [26], showing an ideal correlation between the sizes and supramolecular interactions of all three components.

The study of parent cluster/CD and POM/CD systems is a key point in the preparation of three-component systems. Here, we present our data on the interaction of chloride molybdenum and tungsten clusters with γ-CD.

### 2.2. Synthesis and Structural Characterization of $[M_6Cl^i{}_8Cl^a{}_6]^{2-}/\gamma\text{-CD}$ Complexes

$(H_3O)_2[M_6Cl^i{}_8Cl^a{}_6]$ (M = Mo, W) clusters are stable only in concentrated hydrochloric acid (12 M). Addition of water to these solutions leads invariably to hydrolysis of the compounds with the precipitation of insoluble solids. Sodium salts $Na_2[M_6Cl^i{}_8Cl^a{}_6]$ (obtained by same method reported for other halide clusters [19,21]) are soluble in water, but solutions of the complexes turn black in less than a minute and lead to the formation of brown/black precipitates, indicating complete destruction of the cluster compounds. To avoid the hydrolysis of compounds, it was required to use mixture of ethanol and water. The addition of 5 mL water with $HCl_{conc}$ (50/50 vol.%) solution of γ-CD to 5 mL ethanol solution of cluster compounds $(H_3O)_2[M_6Cl^i{}_8Cl^a{}_6]$ (molar ratio of 3 γ-CD:1 cluster) resulted in precipitation of yellow powders $(H_3O)_2\{[M_6Cl^i{}_8Cl^a{}_6]@(\gamma\text{-CD})_2\}\cdot(\gamma\text{-CD})\cdot15H_2O$ (M = Mo, noted as **Mo₆Cl₁₄@2CD·CD**, and M = W, noted as **W₆Cl₁₄@2CD·CD**), as confirmed by elemental analysis. Important to note, that using the lower amount of γ-CD did not lead to the precipitation of any powders from solution. After separation of the powders from the solutions, the latter still remained colored and, at rest, crystals of inclusion compounds **Mo₆Cl₁₄@2CD·2Mo₆Cl₁₄** or **W₆Cl₁₄@2CD·0.5W₆Cl₁₄** were formed. These structures were confirmed by single-crystal X-ray diffraction analysis (SCXRD, discussed below). An increase in the amount of γ-CD in reaction with **Mo₆Cl₁₄** promoted the precipitation of the same supramolecular assembly **Mo₆Cl₁₄@2CD·CD**, while crystallization from mother liquor resulted in crystals of this compound suitable for SCXRD. The powder X-ray diffraction patterns of the precipitated **M₆Cl₁₄@2CD·CD** from the reaction mixtures were in good agreement with the patterns calculated from the SCXRD data for **Mo₆Cl₁₄@2CD·CD** (Figure S1). The IR spectra of **M₆Cl₁₄@2CD·CD** contained almost unchanged vibrations of the cyclodextrin (Figure S5). The same effect has also been reported for strong supramolecular assemblies of CDs with metal clusters [19–22]. Regardless of the presence of inclusion compounds in the structures, all obtained compounds (both powders and crystals) were unstable in aqueous solutions for more than 2 min.

According to SCXRD, compound $(H_3O)_2\{[Mo_6Cl^i{}_8Cl^a{}_6]@(\gamma\text{-CD})_2\}\cdot2((H_3O)_2[Mo_6Cl^i{}_8Cl^a{}_6])\cdot15H_2O$ (**Mo₆Cl₁₄@2CD·2Mo₆Cl₁₄**) contains supramolecular adduct $\{[Mo_6Cl^i{}_8Cl^a{}_6]@(\gamma\text{-CD})_2\}^{2-}$ co-crystallized with $(H_3O)_2\{[Mo_6Cl^i{}_8Cl^a{}_6]$, resulting in the cluster:CD ratio = 3:2. The inclusion compound presents a typical organization of $[Mo_6Cl^i{}_8Cl^a{}_6]^{2-}$ into two cyclodextrins with participation of secondary rim of CDs. The equatorial plane of the cluster defined by four terminal chloro ligands is parallel to the cyclodextrin cavity, while the remaining two chloro ligands are located inside the cavities (conformational host–guest organization known in the literature as β-form in equivalent clusters with $X^i$ = Br [19,21]) (Figure 1a,b). The cluster inside the CD cavities is disordered in three positions due to statistical rotation around the axis perpendicular to the equatorial plane. In this case, the main interactions of the cluster with cyclodextrins involved hydrogen bonds between terminal chloro ligands and the H3 protons (β position relative to the anomeric position) of glucopyranose fragments (C–H⋯Cl distances of

2.29–2.98 Å depending on disordered position, Figure 1a,b). Two co-crystallized CD-free $[Mo_6Cl^i_8Cl^a_6]^{2-}$ clusters were located close to the primary faces of the cluster-containing CD to form a complex five-component assemblies $Mo_6$-CD@$Mo_6$@CD-$Mo_6$ (Figure 1c,d). Such a combination was interconnected by hydrogen bonds between terminal Cl-ligands of CD-free cluster and H2 protons from γ-CD of neighboring assemblies (C–H⋯Cl distances of 2.59–2.96 Å) or between OH-groups of γ-CDs (O⋯O distances of 2.66–2.71 Å) to form the 3D structure. Solvate water molecules also contributed to the 3D organization and were located in the cavities between assemblies.

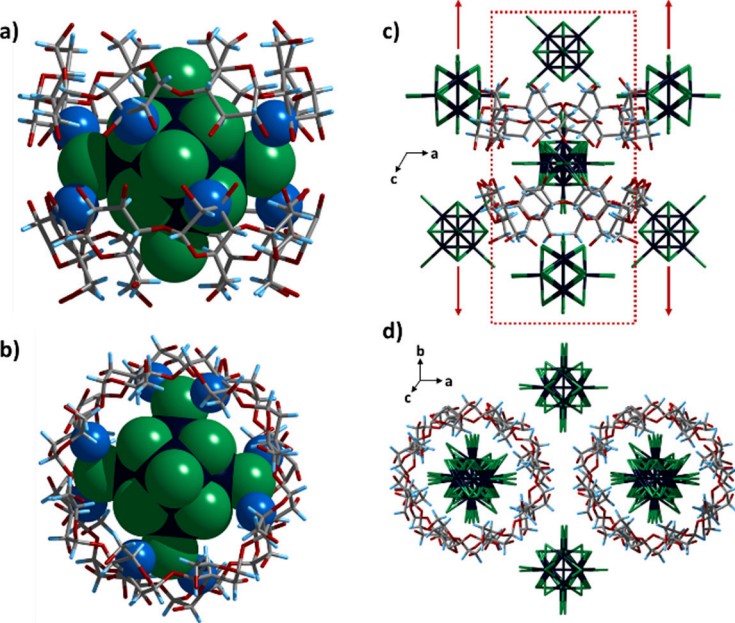

**Figure 1.** Selected fragments of crystal structure of **$Mo_6Cl_{14}$@2CD·2$Mo_6Cl_{14}$**. Side (**a**) and top (**b**) view of inclusion compound $\{[Mo_6Cl^i_8Cl^a_6]@(γ\text{-CD})_2\}^{2-}$. Highlight showing the close environment of the CD-free clusters: side (**c**) and top (**d**) view. Color code: Mo—dark blue, Cl—green, O—red, C—gray, H—light blue, H involved in hydrogen bonding—space-filling blue, $Mo_6$-CD@$Mo_6$@CD-$Mo_6$ complex—red frame. Red arrows indicate growing directions of omitted $Mo_6$-CD@$Mo_6$@CD-$Mo_6$ complex. Disordering of cluster is omitted for clarity in pictures (**a,b**).

Under the same synthesis conditions, the tungsten-containing cluster led to another crystalline product, consistent with formula $(H_3O)_2\{[W_6Cl^i_8Cl^a_6]@(γ\text{-CD})_2\}·0.5((H_3O)_2[W_6Cl^i_8Cl^a_6])·$ $15H_2O$ (noted **$W_6Cl_{14}$@2CD·0.5$W_6Cl_{14}$**). Structural analysis of this compound also revealed a host–guest complex $\{[W_6Cl^i_8Cl^a_6]@(γ\text{-CD})_2\}^{2-}$, co-crystallized with isolated cluster complex (cluster:CD ratio = 3:4). However, the position of the cluster complex within the cyclodextrins differed from that observed in **$Mo_6Cl_{14}$@2CD·2$Mo_6Cl_{14}$** by the rotation of the cluster by 45° (described in the literature as the α-form in equivalent clusters with $X^i$ = I [19]). Hydrogen bonds involving terminal chloro ligands with H3 protons (C–H⋯Cl distances of 2.68–2.79 Å) were depicted, reinforced by additional hydrogen bonds with H5 protons (C–H⋯Cl distances of 2.73–2.86 Å) (Figure 2a,b). The 3D packing was mainly achieved through hydrogen bonds involving terminal chloro ligands of the CD-free cluster and H4 protons of vicinal γ-CDs (C–H⋯Cl distances of 2.78–2.80 Å) (Figure 2c). Each CD-free cluster interacted with four inclusion compounds, while each host–guest complex interacted with only two CD-free clusters. Furthermore, solvate water molecules were involved in 3D organization and located in the available space between assemblies.

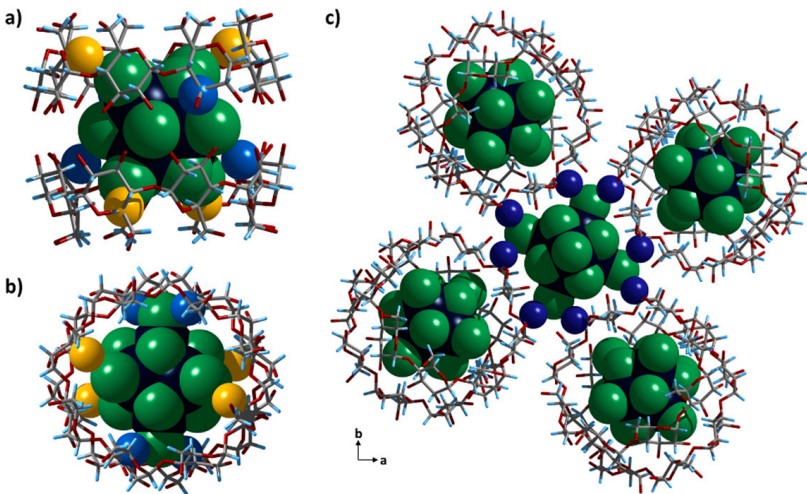

**Figure 2.** Selected fragments of crystal structure of $W_6Cl_{14}@2CD \cdot 0.5W_6Cl_{14}$. Side (**a**) and top (**b**) view of inclusion compound $\{[W_6Cl^i_8Cl^a_6]@(\gamma\text{-CD})_2\}^{2-}$. Highlight of the packing involving CD-free clusters and its four neighboring moieties. (**c**). W—dark blue, Cl—green, O—red, C—gray, H—light blue, H3, H5 and H4 involved in hydrogen bonding—space-filling blue, orange and indigo, respectively.

Increasing the amount of cyclodextrin in the synthesis led to the formation of a compound $(H_3O)_2\{[Mo_6Cl^i_8Cl^a_6]@(\gamma\text{-CD})_2\} \cdot (\gamma\text{-CD}) \cdot 15H_2O$, $Mo_6Cl_{14}@2CD \cdot CD$ differing by its final cluster:CD ratio = 1:3. In the crystal structure, the main supramolecular host–guest adduct $\{[Mo_6Cl^i_8Cl^a_6]@(\gamma\text{-CD})_2\}^{2-}$ co-crystalizes with additional $\gamma$-CD. Cluster in the $\gamma$-CDs appeared disordered over three positions, one consistent with the $\beta$ host–guest conformational form and the other two with the $\alpha$-forms in ratio 0.7:0.15:0.15, respectively. The 3D packing was ensured by hydrogen bonds between primary faces of adjacent $\gamma$-CDs leading to the tubular "bamboolike" structures $-\{CD@Mo_6@CD\text{-}CD\}_n-$ differing from the sequence previously observed $-\{\text{-}CD\text{-}CD@M_6@CD\text{-}CD@M_6@CD\text{-}CD\}_n-$ with other CD-cluster structures [19–21] (Figure 3). The water molecules and the $H_3O^+$ cations were distributed inside large voids between the tubular assemblies.

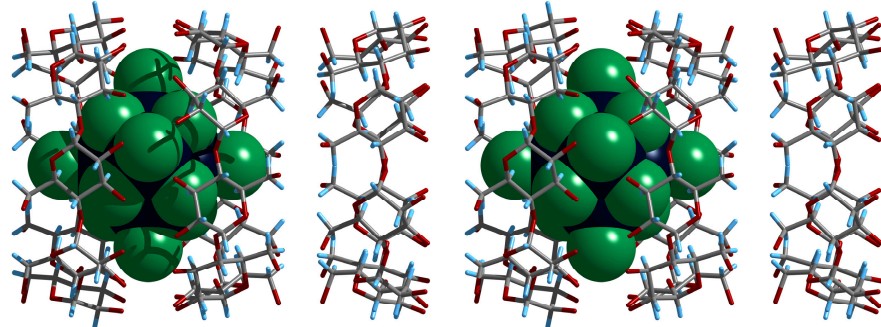

**Figure 3.** Selected fragment of the crystal structure of $Mo_6Cl_{14}@2CD \cdot CD$. Mo—dark blue, Cl—green, O—red, C—gray, H—light blue. Disordering of the cluster is omitted for clarity.

Structural analysis of these reported structures in comparison with those from the literature dealing with $[M_6X^i_8Cl^a_6]^{2-}$ ($X^i$ = Br, I) complexes [19,21] indicated negligible differences in hydrogen bonds between clusters and $\gamma$-CDs. However, the chloro-containing cluster retained some degree of freedom inside the $\gamma$-CD cavities that allowed 45° cluster rotation and resulted in the $\alpha$- and $\beta$-conformational host–guest forms. Therefore, the binding strength of the guest cluster to cyclodextrin was directly related to host–guest matching. Indeed, the cluster size depended mainly on the nature of the inner ligand $X^i$, which increased progressively in the halogeno series from Cl to I. In the presence of the largest iodide-containing clusters (regardless of the metal M = Mo or W), the $\alpha$-form

was favored, while the β-form stabilized when the cluster contained the smaller bromo ligand. The situation appears more intricate for the smallest chloro ligands, where both configurations were observed depending on the nature of the metal M, but also on the cluster:CD ratio found in the solid network. Such a situation was probably due to a very poor host–guest size matching, which does not allow one conformation to be stabilized rather than another. Then, it resulted in the disorder of the $[M_6Cl_{14}]^{2-}$ cluster within the CD as demonstrated previously with rhenium-based clusters [56].

### 2.3. Solution Studies

As the compounds $\mathbf{Mo_6Cl_{14}}$ and $\mathbf{W_6Cl_{14}}$ or their supramolecular systems are not stable in water solution, we decided to carry out $^1H$ NMR titration experiment in $D_2O$-$CD_3OD$ mixed solvent conditions to obtain information about the encapsulation process in such a condition (Figure 4). The experiment was performed on the molybdenum chloride cluster as a representative system. Increasing the number of clusters in a solution of fixed CD concentration resulted in only a moderate downfield signal shift in the NMR spectra for the H3, H5 and H4 protons (the largest shifts are smaller than 0.04 ppm). These protons were the only ones involved in the supramolecular organization of the crystalline structures described above. It is also important to note that the signal shift was practically negligible and much weaker than that observed for other molybdenum clusters $[Mo_6X^i{}_8Cl^a{}_6]^{2-}$ with $X^i$ = Br or I, ($\Delta\delta\sim$ 0.4–0.5 ppm), which indicated the lack of interaction of $\mathbf{Mo_6Cl_{14}}$ and $\gamma$-CD in $D_2O/CD_3OD$ solution.

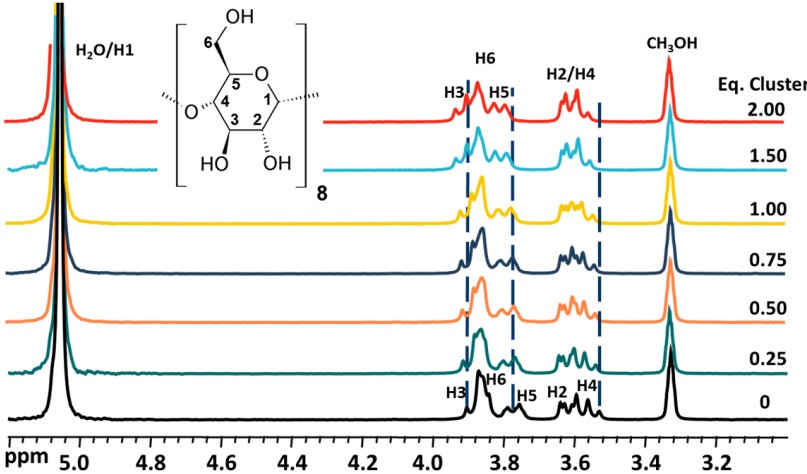

**Figure 4.** $^1H$ NMR spectra from the titration of 2 mM $\gamma$-CD solution with the $\mathbf{Mo_6Cl_{14}}$ in $D_2O/CD_3OD$ (50/50 vol.%) mixture with 5 vol.% of $DCl_{conc}$.

Cluster size (362, 395 and 447 $Å^3$ for X = Cl, Br, and I in $[Mo_6X^i{}_8Cl^a{}_6]^{2-}$, respectively), which affected both host–guest matching and specific-ion effects (chaotropic effect, i.e., water-structure breaking effect [28]), seemed to play an important role in the interaction with $\gamma$-CD. Furthermore, the smallest cluster in the halogeno-metal series exhibited the highest charge density, leading to the decrease in the chaotropic effect of the ion, resulting in weaker interaction in solution with $\gamma$-CD. Moreover, in the halogeno-cluster series, the polarizability was expected to increase following the trend $Cl^i < Br^i < I^i$. In the presence of highly polarizable clusters, the solvation properties of water molecules were strongly affected, inducing water structure breaking effect [17,28]. Although the supramolecular interactions appeared similar in the solid-state for the binary systems, the variation in the chaotropic effect gave consistent explanations upon contrasted behaviors in the stability of inclusion compounds in solution.

### 2.4. Physicochemical Properties

A typical property of halide molybdenum and tungsten cluster complexes is luminescence in the visible and near-IR regions. In this work, the luminescence in the solid state of the powder samples of $M_6Cl_{14}$@2CD·CD in comparison with the CD-free initial complexes were studied (Figure 5, Figures S2 and S3 and Table 1). Upon inclusion in γ-CD, the luminescence profile of the molybdenum complex practically did not change and a slight decrease in the quantum yield was observed. On the other hand, the luminescence of the tungsten cluster appeared to be almost canceled. The observed luminescence quenching could be explained by the close contacts of clusters with γ-CDs, and/or by the presence of a large number of crystallization water molecules or trapped molecular oxygen located in the free available space between the inclusion compounds. The luminescence spectra of $Mo_6Cl_{14}$ and $Mo_6Cl_{14}$@2CD·CD could be fitted by two Voight components [57] (Figure S4 and Table S1). As can be seen from the obtained data, the maxima of the two components, as well as their contribution, practically did not change during the formation of the inclusion compound, indicating a weak influence of the cyclodextrin on the exited states of the molybdenum cluster.

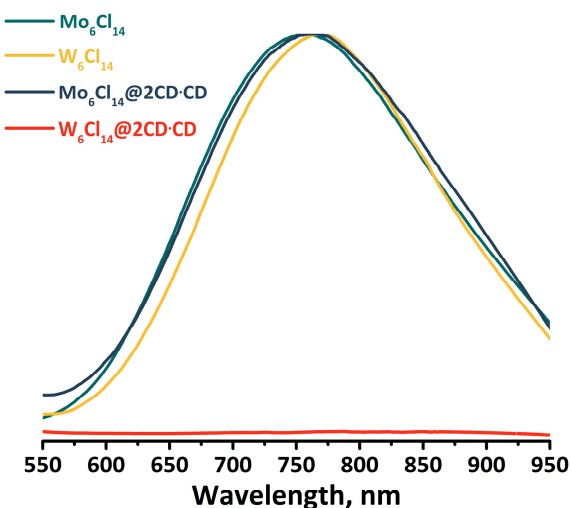

**Figure 5.** Normalized and smoothed emission spectra of compounds in solid state.

**Table 1.** Main photophysical characteristics of clusters $M_6Cl_{14}$ and inclusion compounds $M_6Cl_{14}$@2CD·CD in solid state. λ—maxima of the luminescence bands, τ—emission lifetime and Φ—the luminescence quantum yield.

| Compound | λ, nm | τ (A), μs | $\tau_0$[a], μs | Φ |
|---|---|---|---|---|
| $Mo_6Cl_{14}$ | 750 | $\tau_1 = 30$ (0.29) <br> $\tau_2 = 60$ (0.71) | 55 | 0.10 |
| $Mo_6Cl_{14}$@2CD·CD | 750 | 61 | 61 | 0.08 |
| $W_6Cl_{14}$ | 760 | $\tau_1 = 12$ (0.04) <br> $\tau_2 = 3.5$ (0.96) | 4.6 | 0.05 |
| $W_6Cl_{14}$@2CD·CD | – | – | – | – |

[a] $\tau_0 = (A_1 \cdot \tau_1^2 + A_2 \cdot \tau_2^2)/(A_1 \cdot \tau_1 + A_2 \cdot \tau_2)$.

## 3. Materials and Methods

### 3.1. Materials

$(H_3O)_2[M_6Cl^i_8Cl^a_6] \cdot nH_2O$ (M = Mo, n = 6, noted as $Mo_6Cl_{14}$, M = W, n = 7, noted as $W_6Cl_{14}$) were synthesized as described in the literature [58,59]. Other reagents and solvents were commercially available and used without additional purification.

### 3.2. Methods

Fourier Transform Infrared (FTIR) spectra (Figure S5) were obtained on a 6700 FT-IR Nicolet spectrophotometer in the range 400–4000 cm$^{-1}$. Thermogravimetric analysis was performed on Mettler Toledo TGA/DSC 1, STARe System apparatus under oxygen flow (50 mL·min$^{-1}$) at a heating rate of 5 °C·min$^{-1}$ up to 700 °C. $^1$H NMR spectra were measured in $D_2O/CD_3OD$ (50/50 vol/vol) with 5 vol.% of DCl at 27 °C on a Bruker Avance 400 MHz. Chemical shifts were referenced to TMS. Energy-dispersive X-ray spectroscopy (EDS) measurements were performed using a SEM-FEG (Scanning Electron Microscope enhanced by a Field Emission Gun) equipment (JSM 7001-F, Jeol). Elemental CHNS analysis was performed on a EuroVector EA3000 CHNS analyzer. X-ray powder diffraction (XRPD) patterns were recorded on a Shimadzu XRD 7000S diffractometer (Shimadzu, Kyoto, Japan) (Cu K$\alpha$ radiation, graphite monochromator, and silicon plate as an external standard). The absolute quantum yields were measured using a C9920–03 Hamamatsu system equipped with a 150 W xenon lamp, a monochromator, an integrating sphere and a red-NIR sensitive PMA-12 detector. Lifetime measurements and TRPL mapping were performed using a picosecond laser diode (Jobin Yvon deltadiode, 375 nm) and a Hamamatsu C10910-25 streak camera mounted with a slow single sweep unit. Signals were integrated on a 30 nm bandwidth. Fits were obtained using ORIGIN software and the goodness of fit was judged by the reduced $\chi^2$ value and residual plot shape (Figure S2). The luminescence spectra were smoothed by the Savitzky–Golay filter (points of window—250; polynomial order—2) to increase the signal-to-noise ratio [60].

Single Crystal X-ray Diffraction Analysis

Crystals of compounds were selected from reaction mixture and glued in paratone oil. X-ray intensity data for **Mo$_6$Cl$_{14}$@2CD·CD** were collected at 200 K on a Bruker D8 VENTURE diffractometer equipped with a PHOTON 100 CMOS bidimensional detector using a high brilliance IμS microfocus X-ray Mo K$_\alpha$ monochromatized radiation ($\lambda$ = 0.71073 Å). SAINT V7.53a was used for data reduction. The substantial redundancy in data allowed a semi-empirical absorption correction (SADABS V2.10) to be applied, on the basis of multiple measurements of equivalent reflections. Using Olex2 [61], the structure was solved with the ShelXT [62] structure solution program using Intrinsic Phasing and refined with the ShelXL [63] refinement package using Least Squares minimization. The remaining non-hydrogen atoms were located from Fourier differences and were refined with anisotropic thermal parameters. Positions of the hydrogen atoms belonging to the cyclodextrins were calculated.

The diffraction data for **Mo$_6$Cl$_{14}$@2CD·2Mo$_6$Cl$_{14}$** and **W$_6$Cl$_{14}$@2CD·0.5W$_6$Cl$_{14}$** were collected on a New Xcalibur (Agilent Technologies) diffractometer with MoK$_\alpha$ radiation ($\lambda$ = 0.71073 Å) by doing $\varphi$ scans of narrow (0.5°) frames at 130 K. Absorption correction was done empirically using SCALE3 ABSPACK (CrysAlisPro, Agilent Technologies, Version 1.171.37.35, release 13 August 2014 CrysAlis171 .NET). Structure was solved by direct method with SHELXT [62] and refined by full-matrix least-squares treatment against $|F|^2$ in anisotropic approximation with SHELX 2017/1 [63] in ShelXle program [64]. Hydrogen atoms were refined in geometrically calculated positions.

The crystals of **Mo$_6$Cl$_{14}$@2CD·2Mo$_6$Cl$_{14}$** are complicated inversion twins with random domains combination. The solution was possible only in *P*1 space group giving a favor to refine both CD molecules and disordered octahedral clusters. This disordering opens a nature of domain fusing resulted in violation of the crystal structure due to the displacement of clusters from the positions of the tetragonal crystal system.

The structure of **W$_6$Cl$_{14}$@2CD·0.5W$_6$Cl$_{14}$** contains solvent free voids. Full complex composition was found based on analytical data.

Crystallographic data for single-crystal X-ray diffraction studies are summarized in Table S2. The crystallographic data have been deposed in the Cambridge Crystallographic Data Centre under the deposition codes CCDC 2223838 (**W$_6$Cl$_{14}$@2CD·0.5W$_6$Cl$_{14}$**), 2223839 (**Mo$_6$Cl$_{14}$@2CD·2Mo$_6$Cl$_{14}$**), 2223840 (**Mo$_6$Cl$_{14}$@2CD·CD**).

*3.3. Synthetic Procedures*

3.3.1. Synthesis of $(H_3O)_2\{[Mo_6Cl^i_8Cl^a_6]@(\gamma\text{-}CD)_2\}\cdot(\gamma\text{-}CD)\cdot15H_2O$ ($Mo_6Cl_{14}$@2CD·CD)

$\mathbf{Mo_6Cl_{14}}$ (100 mg, 0.082 mmol) was dissolved in mixture of 5 mL of ethanol and 1 mL of $HCl_{conc}$ under heating. Then, $\gamma$-cyclodextrin (320 mg, 0.246 mmol) was dissolved in mixture of 2.5 mL of water and 2.5 mL of $HCl_{conc}$ and slowly added to ethanol solution of cluster. The powdered product $\mathbf{Mo_6Cl_{14}}$**@2CD·CD** was separated from the solution, washed with an ethanol, and dried in air. Yield 250 mg (58%) based on cluster complex. Anal. Calcd for $C_{144}H_{276}Cl_{14}Mo_6O_{137}$: C, 32.81; H, 5.28. Found: C, 33.2; H, 5.5. EDS shows Mo:Cl ratio = 6:14.1. TGA revealed a weight loss of about 5.2% from 50 to 130 °C (the calculated weight loss of 15 $H_2O$ is 5.1%).

The reaction mixture after separation of $\mathbf{Mo_6Cl_{14}}$**@2CD·CD** was left to stand in closed vial for 2 days resulting in the formation of crystals of compound $(H_3O)_2\{[Mo_6Cl^i_8Cl^a_6]@(\gamma\text{-}CD)_2\}\cdot2((H_3O)_2[Mo_6Cl^i_8Cl^a_6])\cdot15H_2O$ ($\mathbf{Mo_6Cl_{14}}$**@2CD·2Mo₆Cl₁₄**). Yield 30 mg. Anal. Calcd for $C_{96}H_{208}Cl_{42}Mo_{18}O_{101}$: C, 18.61; H, 3.38. Found: C, 18.7; H, 3.1. EDS shows Mo:Cl ratio = 6:14.0.

Increasing amount of $\gamma$-CD (426 mg, 0.328 mmol) in the synthesis results in the precipitation of the same product $\mathbf{Mo_6Cl_{14}}$**@2CD·CD**, while single-crystals of this compound suitable for SCXRD were formed from the reaction mixture upon standing in a closed vial.

3.3.2. Synthesis of $(H_3O)_2\{[W_6Cl^i_8Cl^a_6]@(\gamma\text{-}CD)_2\}\cdot(\gamma\text{-}CD)\cdot15H_2O$ ($W_6Cl_{14}$@2CD·CD)

The compound was obtained using same procedure replacing $\mathbf{Mo_6Cl_{14}}$ by $\mathbf{W_6Cl_{14}}$ (100 mg, 0.057 mmol) and changing amount of $\gamma$-CD (220 mg, 0.170 mmol). Yield 230 mg (70%) based on cluster complex. Anal. Calcd for $C_{144}H_{276}Cl_{14}W_6O_{137}$: C, 29.82; H, 4.80. Found: C, 30.4; H, 5.1. EDS shows W:Cl ratio = 6:13.8. TGA revealed a weight loss of about 5.1% from 50 to 130 °C (the calculated weight loss of 15 $H_2O$ is 4.7%).

The reaction mixture after separation of $\mathbf{W_6Cl_{14}}$**@2CD·CD** was left to stand in closed vial for 2 days resulting in the formation of crystals of compound $(H_3O)_2\{[W_6Cl^i_8Cl^a_6]@(\gamma\text{-}CD)_2\}\cdot0.5((H_3O)_2[W_6Cl^i_8Cl^a_6])\cdot15H_2O$ ($\mathbf{W_6Cl_{14}}$**@2CD·0.5W₆Cl₁₄**). Yield 35 mg. Anal. Calcd for $C_{96}H_{199}Cl_{21}O_{98}W_9$: C, 21.67; H, 3.77. Found: C, 21.7; H, 4.0. EDS shows W:Cl ratio = 9:20.8.

**4. Conclusions**

In conclusion, this work demonstrates that the octahedral chloride clusters are capable of forming host–guest complexes with $\gamma$-CD in the solid state. When cluster and cyclodextrin solutions were mixed, inclusion compounds precipitated in a ratio of 1:3 (a cluster within 2 cyclodextrins co-crystallized with free CD), and crystal structures were obtained from mother liquors with both the same and different ratios of components (3:2 and 3:4 for Mo and W, respectively). Nevertheless, in all structures the fragment $[M_6Cl_{14}]^{2-}$@2CD was observed, confirming the possibility of the formation of inclusion compounds. However, regardless of the structure of the obtained compounds, all adducts were not stable in aqueous solutions due to hydrolysis of the terminal chlorine ligands of the clusters, i.e., cyclodextrin did not prevent/slow down this process. The weak interaction in solution was confirmed by $^1H$ NMR data indicating weaker ion-specific properties of the smaller $[M_6Cl^i_8Cl^a_6]^{2-}$ clusters in a series of complexes with other inner ligands (Br or I). During the formation of inclusion compounds, the luminescence properties of the molybdenum cluster remained practically unchanged, while a complete quenching of the emission was observed for the tungsten compound. The data obtained not only provide new insights into the supramolecular chemistry of octahedral clusters of molybdenum and tungsten and cyclodextrin, but also served as a basis for further combination of cluster complexes with polyoxometalates to create new functional CLUSPOM materials.



**Supplementary Materials:** The supporting information can be downloaded at https://www.mdpi.com/article/10.3390/inorganics11020077/s1, and includes: powder XRD patterns of **Mo$_6$Cl$_{14}$@2CD·CD** and **W$_6$Cl$_{14}$@2CD·CD**, luminescence decay curves of **Mo$_6$Cl$_{14}$**, **Mo$_6$Cl$_{14}$@2CD·CD**, and **W$_6$Cl$_{14}$**, emission spectra of compounds in solid state, deconvolution of the emission spectra of **Mo$_6$Cl$_{14}$** and **Mo$_6$Cl$_{14}$@2CD·CD** in solid state, and FTIR spectra of **Mo$_6$Cl$_{14}$@2CD·CD** and **W$_6$Cl$_{14}$@2CD·CD**.

**Author Contributions:** Conceptualization, A.A.I.; validation, C.F., P.A.A. and M.A.S.; formal analysis, A.A.I., P.A.A., M.H., Y.M. and C.F.; investigation, A.A.I.; resources, S.C.; writing—original draft preparation, A.A.I.; writing—review and editing, P.A.A., C.F., E.C., M.H. and M.A.S.; visualization, A.A.I.; supervision, E.C. and M.A.S.; project administration, E.C.; funding acquisition, M.A.S. All authors have read and agreed to the published version of the manuscript.

**Funding:** The authors gratefully acknowledge financial support from the grant of the President of the Russian Federation (grant no. MD-123.2022.1.3).

**Data Availability Statement:** Crystal structure data can be obtained free of charge from The Cambridge Crystallographic Data Centre via www.ccdc.cam.ac.uk/data_request/cif (accessed on 3 February 2023) or are available on request from the corresponding author.

**Acknowledgments:** Authors thank M.N. Sokolov for helpful discussions. The NIIC team thanks the Ministry of Science and Higher Education of the Russian Federation (no. 121031700321-3, 121031700321-8). Authors also thanks IRP-CNRS CLUSPOM, and LabEx CHARMMMAT (ANR-11-LBX0039-grant), Embassy of France in the Russian Federation for support through the Vernadsky and Metchnikov exchange programs.

**Conflicts of Interest:** The authors declare no conflict of interest.

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
