# Peer review of "Supramolecular Host–Guest Assemblies of [M6Cl14]2–, M = Mo, W, Clusters with γ-Cyclodextrin for the Development of CLUSPOMs"

_inorganics, doi:10.3390/inorganics11020077_

Round 1

Reviewer 1 Report

The paper of Pavel A. Abramov and co-authors is an interesting fundamental work on synthesis a series of Molybdenum and Tungsten halide clusters co-crystallized with γ-cyclodextrin, isolating them in single crystal form amenable for SC X-Ray analysis and determining their crystal structures. Work with cyclodextrins rarely ends with a structural characterization of the reaction products. Solution chemistry methods are commonly used. In this manuscript, the authors managed to isolate single crystals with cyclodextrin, which in the future may lead to additional citation of Inorganics. The article is very neatly written. The work seems very interesting to me and I recommend accepting it for publication in Inorganics.

However, I have some clarifying questions.

Line 59-65: The authors speak of a strong hydrolysis of the clusters. Why then not take an organic solvent that does not contain hydroxo groups from water? (acetonitrile, toluene, benzene, tetrahydrofuran, dimethoxyethane ... and so on - they can be easily dried) Or does cyclodextrin not dissolve in them?

From the text of the manuscript it remains unclear how the correspondence of the single crystal to the powder of the substance was confirmed. Since the authors talk a lot about hydrolysis, powder diffraction data are very scarce.

Line 143: change zero to degree sign at 45⁰

In paragraph 3.2, the authors say that the IR spectrum was taken from the reaction product. Usually, the coordination of ligand leads to a shift of the bands. The IR spectra of the reaction products are absolutely identical to cyclodextrin. How can this be explained? (At first, I thought that maybe the products are hydrolyzed, but then we would see vibrations of hydroxo groups from the products of cluster decomposition)

Is it possible to obtain products by mechanochemistry and compare powder diffraction patterns? It seems to me that this would solve the hydrolysis problem.

Since the article is submitted to the special issue “Advances in Polyoxometalates for Supramolecular Architecture, Biomimetics and Bioapplications”, it is worth giving an explanation, because as far as I understand, this article does not contain polyoxometalates, but there are clusters and cyclodextrins (polyoxo ligands). It might be worth adding some clarifications in the introduction.

Author Response

Referee 1

The paper of Pavel A. Abramov and co-authors is an interesting fundamental work on synthesis a series of Molybdenum and Tungsten halide clusters co-crystallized with γ-cyclodextrin, isolating them in single crystal form amenable for SC X-Ray analysis and determining their crystal structures. Work with cyclodextrins rarely ends with a structural characterization of the reaction products. Solution chemistry methods are commonly used. In this manuscript, the authors managed to isolate single crystals with cyclodextrin, which in the future may lead to additional citation of Inorganics. The article is very neatly written. The work seems very interesting to me and I recommend accepting it for publication in Inorganics.

Answer: Thank you for your positive feedback and relevant suggestions, which we address below.

However, I have some clarifying questions.

Line 59-65: The authors speak of a strong hydrolysis of the clusters. Why then not take an organic solvent that does not contain hydroxo groups from water? (acetonitrile, toluene, benzene, tetrahydrofuran, dimethoxyethane ... and so on - they can be easily dried) Or does cyclodextrin not dissolve in them?

Answer: Cyclodextrin is only soluble in water, DMSO and DMF. It is also important to note that water plays an important role in the interaction of clusters with cyclodextrin. Changing the solvent may have a negative impact on the possibility of forming inclusion compounds.

From the text of the manuscript it remains unclear how the correspondence of the single crystal to the powder of the substance was confirmed. Since the authors talk a lot about hydrolysis, powder diffraction data are very scarce.

Answer: Thank you for important advice. According to powder X-ray diffraction analysis the powder samples correspond to crystalline phase of cluster@2CD with co-crystallized CD (noted Mo6Cl14@2CD·CD). These data as well as discussion were added to manuscript.

Line 143: change zero to degree sign at 45⁰

Answer: The correction was done.

In paragraph 3.2, the authors say that the IR spectrum was taken from the reaction product. Usually, the coordination of ligand leads to a shift of the bands. The IR spectra of the reaction products are absolutely identical to cyclodextrin. How can this be explained? (At first, I thought that maybe the products are hydrolyzed, but then we would see vibrations of hydroxo groups from the products of cluster decomposition)

Answer: According to our knowledge about supramolecular systems of rhenium, niobium, molybdenum or tungsten clusters with cyclodextrins, formation of inclusion compounds does not affect the IR-spectrum of both cluster and cyclodextrin. In this case, the IR spectrum of the compounds is mainly represented by cyclodextrin vibrations. The vibration bands related to cluster complexes are in the low-frequency regions.

Is it possible to obtain products by mechanochemistry and compare powder diffraction patterns? It seems to me that this would solve the hydrolysis problem.

Answer: We conducted a test experiment and, according to powder XRD data, no inclusion compounds are formed in this system. We assume that water molecules and the specific solvation properties of clusters play an important role in the formation of inclusion compounds.

Since the article is submitted to the special issue “Advances in Polyoxometalates for Supramolecular Architecture, Biomimetics and Bioapplications”, it is worth giving an explanation, because as far as I understand, this article does not contain polyoxometalates, but there are clusters and cyclodextrins (polyoxo ligands). It might be worth adding some clarifications in the introduction.

Answer: An additional discussion was added.

Reviewer 2 Report

In general, the authors did an excellent job on this research, and the manuscript is written logically, with the overall claim supported by the results. The article's structure is simple, with a broad discussion on stabilizing halide metal clusters within Host-Guest assemblies with cyclodextrins. 

The structural aspects of the assemblies are well discussed, highlighting the rationale for the supramolecular interactions between both components. Unfortunately, the interactions between the two components are not strong enough to provide the intended halide metal clusters stabilization. 

I am not concerned about the scientific part, and the current study results are exciting and well-discussed. Nevertheless, it would add value to the present manuscript if the authors could provide a paragraph with the most relevant literature containing the best supramolecular methodologies to protect and which compounds provide the strongest supramolecular assemblies with halide metal clusters.

Furthermore, it would be interesting to understand if the fluorescence studies could provide more insights into these interactions. Is it possible to calculate any affinity constants from the fluorescence studies?  

Thank you.

Author Response

Referee 2

In general, the authors did an excellent job on this research, and the manuscript is written logically, with the overall claim supported by the results. The article's structure is simple, with a broad discussion on stabilizing halide metal clusters within Host-Guest assemblies with cyclodextrins. The structural aspects of the assemblies are well discussed, highlighting the rationale for the supramolecular interactions between both components. Unfortunately, the interactions between the two components are not strong enough to provide the intended halide metal clusters stabilization.

Answer: Thank you for your positive feedback and relevant suggestions, which we address below.

I am not concerned about the scientific part, and the current study results are exciting and well-discussed. Nevertheless, it would add value to the present manuscript if the authors could provide a paragraph with the most relevant literature containing the best supramolecular methodologies to protect and which compounds provide the strongest supramolecular assemblies with halide metal clusters.

Answer: Additional information was added to the Introduction part.

Furthermore, it would be interesting to understand if the fluorescence studies could provide more insights into these interactions. Is it possible to calculate any affinity constants from the fluorescence studies?

Answer: If it were possible to study the luminescence properties of clusters in solution with the addition of different amounts of cyclodextrin, then the data obtained would make it possible to extract the formation constants of inclusion compounds. In this case, the obtained luminescence data in solid state do not provide such information. Studies of luminescence in solution have not been carried out due to the low stability of the resulting compounds.

Reviewer 3 Report

This manuscript reports hostguest assemblies of the hexametal cluster [M6Cl14]2 (M = Mo(II) or W(II)) and γ-cyclodextrin. Single-crystal X-ray diffraction analyses revealed encapsulation of the cluster in two γ-cyclodextrin. Hostguest interactions are quite weak and are not observed in the solution phase. The authors claim that a difference between the present system and their previous ones, which have exhibited strong interactions by using [M6X8Cl6]2 (M = Mo(II) or W(II), X = Br or I), arises from the size difference of the clusters. The reviewer agrees their claim. If it is true, however, the authors should test β-cyclodextrin. A pore-size difference between β- (0.70 nm) and γ-cyclodextrin (0.88 nm) matches a difference of the cluster sizes (distance between capping ligands plus twice of ion radii of halide) of [W6Cl14]2 (~0.97 nm) and [W6I8Cl6]2 (~1.13 nm) very well. Furthermore, this manuscript includes several serious issues which must be solved (see below). Thus, the reviewer suggests improvement of the manuscript before the publication.

Quality of crystallographic data is too low to be published. They include enormous serious alerts which are not avoidable by disorder of the cluster unit. The counter cations are also gone by employment of H3O+ salt of the clusters. The authors can employ Na+ salt since both H3O+ and Na+ salts of the clusters are unstable in pure water and they use ethanol/water mixture. An addition of NaCl may enhance the stability if Na+ salt is not stable in ethanol/water.

Emission data are not publishable since 1) the authors have performed non-theoretical smoothing of the spectrum, 2) nature of the samples is not clear (what means “powder samples” (page 6 line 1)? the clusters are really encapsuled in these samples?), 3) emission maximum wavelength of [W6Cl14]2 (760 nm) is shorter than that reported (~800 nm), and 4) emission lifetimes are much shorter than those reported by Kitamura et al. (>100 μs in crystalline phase, Bull. Chem. Soc. Jpn. 2017, 90(10), 1164.). 

Author Response

Referee 3

This manuscript reports host–guest assemblies of the hexametal cluster [M6Cl14]2− (M = Mo(II) or W(II)) and γ-cyclodextrin. Single-crystal X-ray diffraction analyses revealed encapsulation of the cluster in two γ-cyclodextrin. Host–guest interactions are quite weak and are not observed in the solution phase. The authors claim that a difference between the present system and their previous ones, which have exhibited strong interactions by using [M6X8Cl6]2− (M = Mo(II) or W(II), X = Br or I), arises from the size difference of the clusters. The reviewer agrees their claim. If it is true, however, the authors should test β-cyclodextrin. A pore-size difference between β- (0.70 nm) and γ-cyclodextrin (0.88 nm) matches a difference of the cluster sizes (distance between capping ligands plus twice of ion radii of halide) of [W6Cl14]2− (~0.97 nm) and [W6I8Cl6]2− (~1.13 nm) very well. Furthermore, this manuscript includes several serious issues which must be solved (see below). Thus, the reviewer suggests improvement of the manuscript before the publication.

Answer: Thank you for your positive feedback and relevant suggestions, which we address below.

Quality of crystallographic data is too low to be published. They include enormous serious alerts which are not avoidable by disorder of the cluster unit. The counter cations are also gone by employment of H3O+ salt of the clusters. The authors can employ Na+ salt since both H3O+ and Na+ salts of the clusters are unstable in pure water and they use ethanol/water mixture. An addition of NaCl may enhance the stability if Na+ salt is not stable in ethanol/water.

Answer: The quality of the X-ray diffraction data and the quality of crystal structure refinement are typical for complicated structures of huge supramolecular associates. The cause of this is in the nature of crystal packing when: i) huge building blocks form crystal packing suppressing all symmetry of smaller molecules chaotically filling an accessible space; ii) a lot disordering due to suppressing of local symmetry; iii) a lot of molecules of crystallization dramatically affect the crystal packing causing poor diffraction limit of the crystals. The crystals of Mo6Cl14@2CD·2Mo6Cl14 are complicated inversion twins with random domains combination. The solution was possible only in P1 space group giving a favor to refine both CD molecules and disordered octahedral clusters. This disordering opens a nature of domain fusing resulted in violation of the crystal structure due to the displacement of clusters from the positions of the tetragonal crystal system.

Based on the experience of crystallization of supramolecular systems of clusters with cyclodextrins, it is often impossible to determine structurally non-cluster cations or anions. They are randomly distributed in cavities between supramolecular assemblies. Using the sodium salts of the clusters may allow one to determine their positions, but in this case, we were unable to obtain crystalline products. In this case, the cation does not affect the stability of cluster complexes in aqueous solutions, but only allows the dissolution of clusters in water. The addition of salts (like NaCl) will slightly affect the rate of hydrolysis.

Emission data are not publishable since 1) the authors have performed non-theoretical smoothing of the spectrum

Answer: The luminescence spectra were smoothed by the Savitzky–Golay filter (points of window – 250; polynomial order – 2) to increase the signal-to-noise ratio. This method is usually applied for fluorescence spectrum data processing. The parameters for smoothing as well as the unsmoothed spectra were added to manuscript and ESI.

2) nature of the samples is not clear (what means “powder samples” (page 6 line 1)? the clusters are really encapsuled in these samples?)

Answer: Powder samples are the product that precipitates from the reaction mixture immediately upon mixing the cluster solution in ethanol with the cyclodextrin solution in water. Under these conditions, both the cluster and the cyclodextrin do not precipitate from solution for a long time without adding the second component. Only the presence of both components in a certain ratio leads to the formation of inclusion compounds, which already indicates the presence of any interactions of compounds. Moreover, according to the powder XRD data, the powder diffractograms correlate with the theoretical one constructed from the single-crystal XRD data for Mo6Cl14@2CD·CD.

3) emission maximum wavelength of [W6Cl14]2− (760 nm) is shorter than that reported (~800 nm)
4) emission lifetimes are much shorter than those reported by Kitamura et al. (>100 μs in crystalline phase, Bull. Chem. Soc. Jpn. 2017, 90(10), 1164.). 

Answer: The luminescence of cluster complexes in a solid often depends on the packing of compounds, cations, solvate molecules, etc. The mentioned article by Prof. Kitamura is devoted to the study of the luminescence of the TBA salt of the tungsten cluster, for which (among other salts) the best photophysical characteristics are most often demonstrated. To the best of our knowledge, there are no published data for (H3O)2[W6Cl14] in the literature. As example for the comparison of luminescence of clusters a few examples can be mentioned: emission maximum of Mo6Cl12, (H3O)2Mo6Cl14, Cs2Mo6Cl14, (Bz3NH)2[Mo6Cl14] and (Bu4N)2Mo6Cl14 are 760 (Ströbele et al., Z. Anorg. Allg. Chem. 2009, 635, 822), 725 (Kozhomuratova et al., Russ. J. Coord. Chem., 2007, 33, 1), 702 (Grasset et al., Adv. Mater. 2008, 20, 143), 730 (Kozhomuratova et al., Russ. J. Coord. Chem., 2007, 33, 213) and 805 (Maverick et al., J. Am. Chem. Soc., 1983, 105, 1878) nm correspondingly; emission maximum of Cs2Mo6Br14 and (Bu4N)2Mo6Br14 are 704 (Grasset et al., Adv. Mater. 2008, 20, 143) and 830 (Maverick et al., J. Am. Chem. Soc., 1983, 105, 1878) nm correspondingly etc. Thus, a significant difference in the photophysical characteristics of the complexes described in the work from the literature ones is primarily due to the different composition of the compounds.

Round 2

Reviewer 3 Report

The manuscript was revised adequately. It is, therefore, publishable after applying minor self-revising: abbreviations without any annotations (e.g., CLUSPOM and ligands in the first paragraph in section 2.1), "D6h" (Line 98, "D" in italic and "6h" in subscript), etc.

Author Response

Referee 3

The manuscript was revised adequately. It is, therefore, publishable after applying minor self-revising: abbreviations without any annotations (e.g., CLUSPOM and ligands in the first paragraph in section 2.1), "D6h" (Line 98, "D" in italic and "6h" in subscript), etc.

Answer: The required corrections were made.